# Transcriptome-wide association study of attention deficit hyperactivity disorder identifies associated genes and phenotypes

Calwing Liao[1,2], Alexandre D. Laporte[2], Dan Spiegelman[2], Fulya Akçimen [1,2], Ridha Joober[3], Patrick A. Dion[2,4] & Guy A. Rouleau[1,2,4]*

Attention deficit/hyperactivity disorder (ADHD) is a common neurodevelopmental psychiatric disorder. Genome-wide association studies (GWAS) have identified several loci associated with ADHD. However, understanding the biological relevance of these genetic loci has proven to be difficult. Here, we conduct an ADHD transcriptome-wide association study (TWAS) consisting of 19,099 cases and 34,194 controls and identify 9 transcriptome-wide significant hits, of which 6 genes were not implicated in the original GWAS. We demonstrate that two of the previous GWAS hits can be largely explained by expression regulation. Probabilistic causal fine-mapping of TWAS signals prioritizes KAT2B with a posterior probability of 0.467 in the dorsolateral prefrontal cortex and TMEM161B with a posterior probability of 0.838 in the amygdala. Furthermore, pathway enrichment identifies dopaminergic and norepinephrine pathways, which are highly relevant for ADHD. Overall, our findings highlight the power of TWAS to identify and prioritize putatively causal genes.

[1] Department of Human Genetics, McGill University, Montréal, QC, Canada. [2] Montreal Neurological Institute, McGill University, Montréal, QC, Canada. [3] Department of Psychiatry, McGill University, Montréal, QC, Canada. [4] Department of Neurology and Neurosurgery, McGill University, Montréal, QC, Canada. *email: guy.rouleau@mcgill.ca

Attention deficit/hyperactivity disorder (ADHD) is a common neurodevelopmental disorder globally affecting 2.5% of adults and 5% of children[1]. The disorder has been shown to be highly heritable and increases risk of substance abuse, suicide, and risk-taking behavior[2]. Brain-imaging studies have identified various different regions, such as the cerebellum and frontal cortex, to be implicated in ADHD[3,4]. Twin studies have estimated the narrow-sense heritability of ADHD to be ~70%, suggesting a strong genetic component is driving the phenotypic variance[5]. Recently, a large-scale genome-wide association study (GWAS) identified 12 loci that were significantly associated with ADHD[6]. Despite the significant success of GWAS in delineating elements that contribute to the genetic architecture of psychiatric disorders, the loci identified are frequently difficult to characterize biologically. Often, these studies associate loci with the nearest gene, which inevitably leads to a bias for longer genes, and may not necessarily accurately depict the locus's real effect. In contrast, transcriptomic studies have allowed for more interpretable biologically relevant results due to their use of disease-relevant cell-types and tissue, as well as the availability of databases detailing the tissue-specific expression[7]. It is also important to denote that transcriptomic studies conducted for brain disorders tend to have small sample size, by comparison to the studies of conditions where disease relevant tissue is more easily obtainable than brain tissue.

Recently, transcriptomic imputation (TI) was developed and is a powerful method to integrate genotype and expression data from large consortia, such as the Genotype-Tissue Expression (GTEx) through a machine-learning approach[7]. This method derives the relationship between genotypes and gene expression to create reference panels consisting of predictive models applicable to larger independent datasets[8]. Ultimately, TI provides the opportunity to increase the ability to detect putative genes with small effect sizes that are associated with a disease.

To identify genetically regulated genes associated with ADHD, we leverage the largest ADHD cohort currently available to conduct a transcriptome-wide association study (TWAS); the cohort consists of 19,099 ADHD cases and 34,191 controls from Europe. Brain-tissue derived TI panels were used, including the 11 brain-relevant tissue panels from GTEx 53 v7 and the CommonMind Consortium (CMC). Here, we show that nine genes reach within tissue panel Bonferroni-corrected significance. We additionally identify three loci and genes that were not previously implicated with ADHD. Through conditional analyses, we demonstrate that several of the genome-wide significant signals from the ADHD GWAS are driven by genetically regulated expression. Gene set analyses of the Bonferroni-corrected TWAS genes have identified relevant pathways, among which dopaminergic neuron differentiation and norepinephrine neurotransmitter release cycle. Additionally, by querying the top eQTLs identified by TWAS in

phenome databases, we identify several phenotypes previously associated with ADHD, such as educational attainment, body mass index (BMI), and maternal smoking around birth. Finally, genetic correlation of the pheWAS traits demonstrate that several Bonferroni-corrected significant correlations with risk-related behaviors, such as increased number of sexual partners and ever-smoking. In conclusion, TWAS is a powerful method that increases statistical power to identify small-effect size in genes associated with complex diseases such as ADHD.

## Results

**Transcriptome-wide significant hits**. To identify genes associated with ADHD, a TWAS was conducted using FUSION and within panel Bonferroni-corrected thresholds[7] (Supplementary Table 1). A total of nine genes were found to be significantly associated with ADHD (Table 1 and Fig. 1). Amongst the signals, six of the genes were not implicated in the original ADHD GWAS and three were previously implicated. To assess inflation of imputed association statistics under the null of no GWAS association, the QTL weights were permuted to empirically determine an association statistic. The majority of genes were still significant after permutation, suggesting their signal is genuine and not due to chance.

**ADHD TWAS loci are driven by expression signals**. Since several of the TWAS hits overlapped with significant ADHD loci, conditional and joint analyses were performed to establish whether these signals were due to multiple-associated features or conditionally independent. It was observed that $AP006621.5$ explains all of the signal at its loci (rs28633403 lead $SNP_{GWAS}$ $P = 4.5E-07$, conditioned on $AP006621.5$ lead $SNP_{GWAS}$ $P = 1$) (Fig. 2a). It was also found that $RNF219$ explains most of the signal (rs1536776 lead $SNP_{GWAS}$ $P = 5.5E-07$, conditioned on $RNF219$ to lead $SNP_{GWAS}$ $P = 5.1E-02$) explaining 0.848 of the variance (Fig. 2b). Conditioning on $MANBA$ completely explained the variance of the loci on chromosome 4 (rs227369 Lead $SNP_{GWAS}$ $P = 1.3E-07$, lead $SNP_{GWAS}$ $P = 1$) (Fig. 2c).

**Several ADHD loci are explained by expression signals**. Similarly, conditioning on the expression of $ELOVL1$, $CCDC24$, and $ARTN$ depending on the panel demonstrates expression-driven signals in a previously implicated ADHD loci (rs11420276 lead $SNP_{GWAS} = 1.1E-12$, when conditioned on $ELOVL1$, $CCDC24$, and $ARTN$ lead $SNP_{GWAS} = 7.1E-04$) explaining 0.774 of the variance (Fig. 2d). $CCDC24$ had a cross-validation $R^2$ of 0.074, $ELOVL1$ with an $R^2$ of 0.015, and $ARTN$ with an $R^2$ of 0.045 in the putamen basal ganglia, and 0.264 in the cerebellar hemisphere. These genes had a less extreme $Z$-score compared to the GWAS SNP, which prompted conditional analysis. For the

### Table 1 Significant TWAS genes for ADHD

| TWAS identified gene | Tissue | Best eQTL | Direction, Z-score | TWAS P-value | Permutation P-value | Implicated in 2019 ADHD GWAS | Previously implicated GWAS loci | Previous GWAS |
|---|---|---|---|---|---|---|---|---|
| implicated genes | | | | | | | | |
| CCDC24 | Cerebellum | rs12741964 | −6.11 | 9.48E−10 | 0.02 | Yes | 1:44184192 | a |
| ARTN | Putamen basal ganglia | rs2906457 | 5.68 | 1.31E−08 | 0.04 | Yes | 1:44184192 | a |
| ARTN | Cerebellar hemisphere | rs223508 | 5.65 | 1.59E−08 | 0.06 | Yes | 1:44184192 | a |
| ELOVL1 | DLPFC | rs1199039 | 5.64 | 1.62E−08 | 0.05 | No | 1:44184192 | a |
| TIE1 | DLPFC | rs3768046 | 5.27 | 1.31E−07 | 0.18 | No | 1:44184192 | a |
| MED8 | DLPFC | rs11210892 | −5.14 | 2.72E−07 | 0.06 | No | 1:44184192 | a |
| MANBA | Cerebellar hemisphere | rs223508 | 5.19 | 2.04E−07 | 0.001 | No | 1:44184192 | a |
| CTC-498M16.4 | Substantia nigra | rs10044618 | −5.42 | 5.91E−08 | 0.07 | No | 5:87854395 | b |
| RNF219-AS1 | Frontal cortex BA9 | rs1410739 | 5.10 | 3.39E−07 | 0.0003 | No | – | – |

[a]ST3GAL3, KDM4A, KDM4A-AS1, PTPRF, SLC6A9, ARTN, DPH2, ATP6V0B
[b]LINC00461, MIR9-2, LINC02060, TMEM161B-AS1

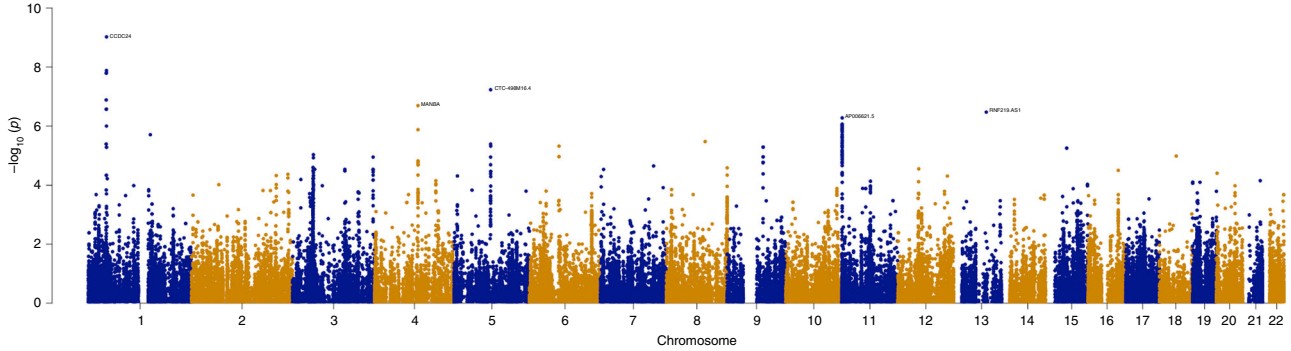

**Fig. 1** Manhattan plot of the transcriptome-wide association study for ADHD ($n = 19,099$ cases and $n = 34,194$ controls). Bonferroni-corrected significant genes are labeled. A significance threshold of $P = 4.97E-07$ was used

| Table 2 Omnibus significant TWAS genes for ADHD | |
|---|---|
| **Gene** | **Omnibus *P*-value** |
| *ARTN* | 3.27E−11 |
| *CCDC24* | 3.90E−10 |
| *LINC00951* | 2.01E−08 |
| *STGAL3* | 3.27E−07 |
| *AP006621.1* | 1.17E−06 |
| *MED8* | 1.60E−06 |
| *CTC-498M16.4* | 2.00E−06 |

previously implicated ADHD GWAS loci at chromosome 5, conditioning on *CTC-498M16.4* explains 0.765 of the variance (rs4916723 lead SNP$_{GWAS}$ $P = 1.8E-08$, lead SNP$_{GWAS}$ $P = 6.4E-03$) (Fig. 2e). The CTC-498M16.4 gene had a cross-validation $R^2$ of 0.056, with a less extreme $Z$-score.

**Omnibus testing reinforces relevance of several genes**. To test for whether an effect was occurring across the different panels, an omnibus test was used. There were seven genes that passed Bonferroni-corrected significance and shown to be associated with ADHD. Interestingly, *CCDC24*, *ARTN*, *AP006621.1*, *CTC-498M16.4*, and *MED8* remained significant. The long non-coding RNA *LINC00951* and *ST3GAL3* did not reach transcriptome-wide significance in the individual panels, but the combined omnibus score increased power to detect a signal (Table 2).

**Fine-mapping of TWAS signals provides evidence of causality**. To prioritize putatively causal genes, FOCUS was used to assign a posterior inclusion probability for genes at each TWAS region and for relevant tissue types. For the genomic locus 3:20091348–3:21643707, *KAT2B* was included in the 90%-credible gene set with a posterior probability of 0.467 in the dorsolateral prefrontal cortex (Table 3). For the genomic loci 5:87390784–5:88891530, *TMEM161B*, *CTC-498M16.4*, and *CTC-498M16.2* were part of the credible set. The highest posterior probability for causality was 0.838 for *TMEM161B* in the amygdala and 0.139 for *CTC-498M16.4* for the hypothalamus. For the locus 16:71054116–16:72934341, *TXNL4B*, *HPR*, *DHODH*, *ZNF23*, *HP*, *IST1*, *DHX38*, and *DDX19A* were included in the credible gene set. However, all the genes had lower posterior inclusion probabilities (Table 3).

**Pathway enrichment**. To understand the biologically relevant pathways from the transcriptome-wide significant hits, pathway

and gene ontology analyses were conducted using Reactome and GO. The genes were grouped into three different clusters based on co-expression of public RNA-seq data ($n = 31,499$) (Supplementary Fig. 2). Several relevant pathways were significantly enriched, such as dopaminergic neuron differentiation (Mann–Whitney *U*-Test, $P = 3.5E-03$), norepinephrine neurotransmitter release cycle (Mann–Whitney *U*-Test, $P = 4.4E-03$), and triglyceride lipase activity (Mann–Whitney *U*-Test, $P = 2.9E-03$) when analyzing all genes together. Interestingly, several relevant cellular regions such as the axon and dendritic shaft were also enriched (Supplementary Table 2).

**Phenome-wide association study**. To understand phenotypes that may be associated or co-morbid with ADHD, a pheWAS was done for each eQTL (Supplementary Table 3). Since most eQTLs were associated with ADHD, we chose to exclude it from Supplementary Table 3 to emphasize the other three top phenotypes per SNP. Several risk-associated phenotypes such as ever-smoker, alcohol intake over 10 years, and maternal smoking around birth were found to be significantly associated with the eQTLs. These phenotypes have previously been implicated as risk factors for ADHD, reaffirming the relevance of the eQTLs.

**Genetic correlation of pheWAS traits**. To determine whether the pheWAS traits were genetically correlated and in which direction, genetic correlation was done between the most recent (as of the writing of this publication) GWAS for each of the phenotypes[6]. Interestingly, there was a strong negative correlation between educational attainment and ADHD (Supplementary Fig. 1). Furthermore, there was a positive correlation with maternal smoking around birth, body mass index, ever smoker, and schizophrenia. Most of these phenotypes, except for maternal smoking were previously implicated in the GWAS paper.

**Discussion**
ADHD is a common disorder that affects millions of people worldwide. While recent GWAS has been successful and identifying risk loci associated with ADHD, the functional significance of these associations continue to remain elusive due to the inability to fine-map to tissue-specific and tissue-relevant genes. Here, we conducted an ADHD TWAS using the summary statistics of over 50,000 individuals from the most recent ADHD GWAS. This approach creates genotype-expression reference panels using public consortia through machine-learning approaches, allowing for imputation and association testing of independent large-scale data[7,8]. We identified nine genes-associated with ADHD risk and different tissue types, localizing to five different regions in the genome. Interestingly, conditional and joint analyses

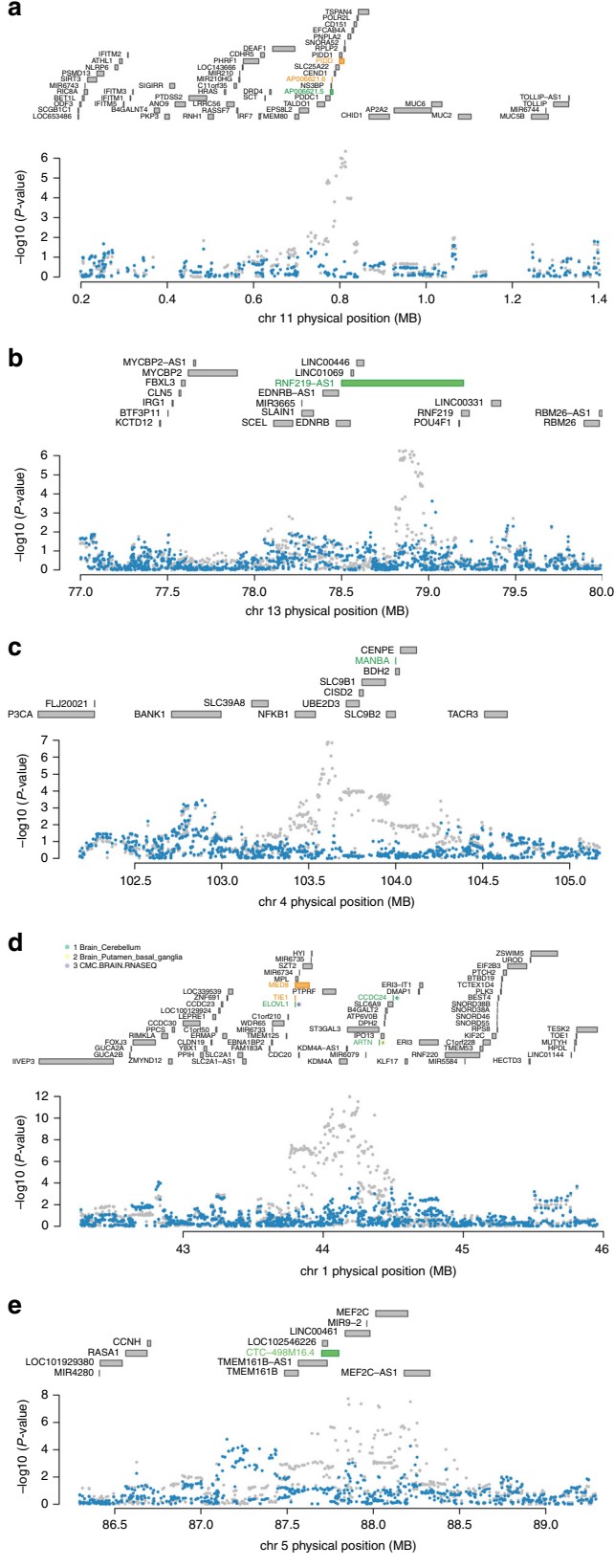

**Fig. 2** Regional association of TWAS hits. **a** Chromosome 11 regional association plot. **b** Chromosome 13 regional association plot. **c** Chromosome 4 regional association plot. **d** Chromosome 1 regional association plot. **e** Chromosome 5 regional association plot. The top panel in each plot highlights all genes in the region. The marginally associated TWAS genes are shown in orange and the jointly significant genes are shown in green. The bottom panel shows a regional Manhattan plot of the GWAS data before (gray) and after (blue) conditioning on the predicted expression of the green genes

GWAS signal. This suggests that there is little residual association signal from the genetic variant in the GWAS locus after accounting for these predicted expression signals. Similarly, *CTC-498M16.4* conditioning also explained a large variance of the GWAS locus. Future studies could interrogate whether expression differences are consistent with these findings. Furthermore, for smaller genes, such as *AP006621.5 gene*, they would normally go unnoticed due to the many larger protein-coding genes nearby. However, our TWAS results demonstrated that the expression of *AP006621.5* fully explained the suggestive ADHD GWAS signal, highlighting the power of TWAS to prioritize genes of interest. Moreover, across all brain tissue types, the significant hits were consistently seen in the following biologically relevant tissue for ADHD: cerebellum, dorsolateral prefrontal cortex, frontal cortex, basal ganglia, and anterior cingulate cortex. These regions are consistent with previously implicated deficit points in the frontal-subcortical catecholamine and dopamine networks for ADHD[9,10]. Furthermore, certain genes were Bonferroni-corrected significant only in certain brain tissue types. For instance, *CCDC24* was significant only in the cerebellum, *ELOVL1*, *TIE1*, and *MED8* were specific to the DLPFC. Since expression regulation may be common across tissue types, it was interesting to not see consistency across panels. For instance, *MED8* had a *P*-value of 2.72E−07 in the DLPFC but a *P*-value of 0.157 in the frontal cortex. Although it may be due to tissue-specificity, it is important to note that it may also be due to panel-specific effects and the quality of the RNA data and panel size from GTEx and CMC. Another example, *ARTN*, may not be brain tissue-specific, since it was significant in the omnibus test and in more than one brain tissue, but instead may be dysregulated at large. However, many TWAS hits tend to be correlated due to co-expression. Causal gene prioritization programs, such as using FOCUS probabilistically help fine-map towards credible genes[11].

Fine-mapping of TWAS hits included *KAT2B* in the credible-set with a posterior probability of 0.467 in the dorsolateral prefrontal cortex. The literature shows that ADHD has been associated with weaker function of the prefrontal cortex compared to healthy individuals[12]. *KAT2B* is a lysine histone acetyltranferase highly expressed in the brain[13]. Previous evidence has suggested that lysine acetylation is importance for brain function and proper development[13]. At another locus, *TMEM161B*, a trans-membrane protein, had the highest posterior inclusion probability of 0.838 in the amygdala. Brain imaging studies have shown that the amygdala has decreased volume in ADHD patients[14]. Genetic variants in the gene have also been previously associated with major depressive disorder (MDD), which is a disorder that is often co-morbid with ADHD[15]. Furthermore, genetic correlation of ADHD and MDD has been shown to have a significant positive genetic correlation[16]. Another gene at this locus, *CTC-498M16.4*, was included in the credible-set for multiple relevant brain tissue types, such as the hypothalamus, hippocampus, and amygdala as well. Although, the posterior inclusion probability was lower for this gene, *CTC-498M16.4*, also

demonstrated that the TWAS expression signals were driving the significance for several previously implicated ADHD loci when conditioned on the top TWAS gene. The multi-gene conditioning of *ELOV1*, *CCDC24*, and *ARTN* led to explained 77.4% of the

**Table 3 Causal posterior probabilities for genes in 90%-credible sets for ADHD TWAS signals with Z-score >|3|**

| Region | Gene | Tissue | TWAS Z | Posterior probability for causality |
|---|---|---|---|---|
| 3:20091348–21643707 | KAT2B | DLPFC | −4.57 | 0.47 |
| 5:87390784–88891530 | TMEM161B | Amygdala | 5.18 | 0.84 |
| 5:87390784–88891530 | CTC-498M16.4 | Hypothalamus | −4.68 | 0.14 |
| 5:87390784–88891530 | CTC-498M16.2 | Hippocampus | −4.69 | 0.08 |
| 5:87390784–88891530 | CTC-498M16.4 | Nucleus accumbens basal ganglia | −4.59 | 0.08 |
| 5:87390784–88891530 | CTC-498M16.4 | Amygdala | −3.98 | 0.06 |
| 16:71054116–72934341 | TXNL4B | Substantia nigra | 4.3 | 0.02 |
| 16:71054116–72934341 | HPR | Hippocampus | 4.04 | 0.01 |
| 16:71054116–72934341 | HPR | Amygdala | 3.99 | 0.01 |
| 16:71054116–72934341 | HPR | Cerebellar hemisphere | 3.92 | 0.01 |
| 16:71054116–72934341 | HPR | Frontal cortex | 3.87 | 0.01 |
| 16:71054116–72934341 | HPR | Anterior cingulate cortex BA24 | 3.85 | 0.01 |
| 16:71054116–72934341 | TXNL4B | Anterior cingulate cortex BA24 | 3.81 | 0.01 |
| 16:71054116–72934341 | HPR | Nucleus accumbens basal ganglia | 3.76 | 0.01 |
| 16:71054116–72934341 | HPR | Substantia nigra | 3.76 | 0.01 |
| 16:71054116–72934341 | HPR | Brain cortex | 3.73 | 0.01 |
| 16:71054116–72934341 | HPR | Cerebellum | 3.67 | 0.01 |
| 16:71054116–72934341 | HP | Anterior cingulate cortex BA24 | 3.65 | 0.01 |
| 16:71054116–72934341 | HP | Caudate basal ganglia | 3.64 | 0.01 |
| 16:71054116–72934341 | TXNL4B | Amygdala | 3.48 | 0.01 |
| 16:71054116–72934341 | TXNL4B | Dorsolateral prefrontal cortex | 3.49 | 0.01 |
| 16:71054116–72934341 | HPR | Caudate basal ganglia | 3.45 | 0.01 |
| 16:71054116–72934341 | DDX19A | Caudate basal ganglia | 1.68 | 0.01 |
| 16:71054116–72934341 | TXNL4B | Caudate basal ganglia | 3.23 | 0.01 |

known as *lnc-TMEM161B-3:2*, a lncRNA was amongst the top prioritized hits for the TWAS, omnibus test, and fine-mapping.

Clustering the TWAS hits into a gene network based on co-expression identified that *ELOVL1*, *TIE1*, and *MED8* were co-expressed, with *ELOVL1* and *MED8* having a stronger co-expression. These hits also were specific to the DLPFC. Similarly, *CCDC24* and *ARTN* clustered together separate from the former three genes and are implicated in cerebellar tissue, despite all six hits resulting from the same locus. It is likely that these two clusters represent two separate hits and the genes within the cluster are simply co-expressed. This suggests that the same locus can have multiple unique TWAS hits across different tissues, and gene clustering could reaffirm if there is a tissue-specific dysregulation[8]. Interestingly, pathway and GO enrichment reinforced several pathways that have previously been reported as biologically relevant. Both dopaminergic and noradrenergic contributions have been implicated in the pathogenesis of ADHD[10]. For the top eQTLs associated with each transcriptome-wide significant gene, many re-occurring phenotypes relevant to ADHD were present, such as ever-smoker and number of sexual partners. A genetic correlation between those available traits from public GWAS data and the most recent ADHD GWAS found inverse correlation for education attainment, consistent with studies on educational outcome with ADHD. Additionally, a genetic correlation for risky behaviors such as ever-smoker and maternal smoking around birth were positively correlated. Maternal smoking has often been suggested to be a risk factor for ADHD[17]. However, the positive genetic correlation could suggest pleiotropy for genetic loci associated with both phenotypes. This would be consistent with pheWAS results showing that some eQTLs were highly associated with both ADHD and smoking. Two recent studies have inquired about ADHD eQTLs and both found overlapping results. In Fahira et al. (2019), the researchers used Sherlock, which is a colocalization method and investigated eQTLs in GTEx, but do not consider the CMC[18]. Next, the researchers used a summary mendelian randomization was done to identify putatively causal genes, however, this method does not consider tissue specificity. In contrast, FOCUS accounts for this[11]. Similarly, Gamazon et al. (2019) has done a multi-tissue analyses in several neuropsychiatric traits, such as ADHD, bipolar disorder, schizophrenia using PrediXcan[19,20]. It focuses primarily on large comparisons between these traits but does not go in as in depth into ADHD. Although, similarly identify several hits overlapping with the results in this paper: *ARTN*, *MED8*, and *TIE1*.

We conclude this study with several caveats and potential follow-up studies. First, TWAS associations could potentially be due to confounding because the gene expression levels that were imputed are derived from weighted linear combinations of SNPs. These SNPs could be included in non-regulatory mechanisms driving the association and risk, ultimately inflating certain statistics. Although the permutation tests and probabilistic fine-mapping used in this study try to protect against these spurious chance events, there is still a possibility of this occurring. Second, a follow-up study will require a large replication cohort, which may be difficult to ascertain, since this current largest GWAS dataset was used in this study. Future studies could investigate the possibility of using gene-risk scores in additional cohorts to validate any findings from this study. Finally, a given gene may have other regulatory features that do not go through eQTLs and still have downstream effect on the trait. Here, we successfully managed to identify several putatively causal genes such as *TMEM161B* and *KAT2B* associated with ADHD. To conclude, TWAS is a powerful statistical method to identify small and large-effect genes associated with ADHD and helps with understanding the molecular underpinning of the disease.

## Methods

**Genotype data**. Summary statistics were obtained through the ADHD Workgroup of the Psychiatric Genomics Consortium (PGC-ADHD)[21]. Details pertaining to participant ascertainment and quality control were previously reported by Demontis et al.[6] The data used in this paper includes only the

European population from the ADHD GWAS ($n = 19,099$ cases and $n = 34,194$ controls).

**Transcriptomic imputation**. TI was done using eQTL reference panels derived from tissue-specific gene expression coupled with genotypic data using panels from FUSION[7]. Here, we used 10 brain tissue panels from GTEx 53 v7 and the CommonMind Consortium (CMC)[22]. A strict Bonferroni-corrected study-wise threshold was used: $P = 4.97E{-}07$ (0.05/100,572) (total number of genes across panels). FUSION was used to conduct the transcriptome-wide association testing. The 1000 Genomes v3 LD panel was used for the TWAS. FUSION utilizes several penalized linear models, such as GBLUP, LASSO, Elastic Net[7]. Additionally, a Bayesian sparse linear mixed model (BSLMM) is used. FUSION computes an out-sample $R^2$ to determine the best model by performing a fivefold cross-validating of every model. After, a multiple degree-of-freedom omnibus test was done to test for effect in multiple reference panels. This test will account for pairwise correlation between functional features. The threshold for the omnibus test was $P = 4.64E{-}06$ (0.05/10,323) (number of genes tested for omnibus).

**Conditionally testing GWAS signals and permutation**. To determine how much GWAS signal remains after the expression association from TWAS is removed, joint and conditional testing was done for genome-wide Bonferroni-corrected TWAS signals using FUSION[7]. The defined regions include only the transcribed region of the genes. Each ADHD GWAS SNP association was conditioned on the joint gene model one SNP at a time. To assess inflation of imputed association statistics under the null of no GWAS association, a permutation test ($n = 100,000$ permutations) was conducted to shuffle the QTL weights and empirically determine an association statistic. Permutation was done for each of the significant loci using FUSION. The loci that pass the permutation test demonstrate levels of heterogeneity captured by expression and are less likely to be co-localization due to chance. It should be noted that the permuted statistic is very conservative and truly causal genes could fail to reject the null due to the QTLs having complex and high linkage disequilibrium.

**Fine-mapping of TWAS associations**. To address the issue of co-regulation in TWAS, we used the program FOCUS (Fine-mapping of causal gene sets) to directly model predicted expression correlations and to give a posterior probability for causality in relevant tissue types[11]. FOCUS identifies genes for each TWAS signal to be included in a 90%-credible set while controlling for pleiotropic SNP effects. The same TWAS reference panels for FUSION were used as in the analysis described above.

**Gene-set analyses**. Due to the stringent Bonferroni-corrected significance, we relaxed the threshold for pathway analyses since Bonferroni-correction assumes independence and genes tend to be correlated due to co-expression. A relaxed nominal Bonferroni-corrected threshold of 0.10 (uncorrected $9.94E{-}07$) was used because co-regulation in TWAS signals violate the independence assumption required for Bonferroni-correction, making it too strict, especially for gene-set enrichment. For gene-set enrichments, more genes will allow for better recapitulation and prioritization of appropriate pathways. Gene clustering was done using the GeneNetwork v2.0 (https://genenetwork.nl) RNA-sequencing database ($n = 31,499$)[23]. This was used because GeneNetwork it identifies co-regulated genes within each pathway, which can help differentiate whether co-regulation is due to proximity to the same eQTL in TWAS or converges independent from different TWAS hits. Briefly, a principal component analysis (PCA) is done on the 31,499 RNA-seq and the eigenvector coefficients for reliable principal components (PC). Co-regulation scores are calculated between genes, which is the correlation between the eigenvector coefficients for each pair. Next, for each reliable PC, it is determined how much it explains each biological pathway, which is defined as the group of genes annotated with the term in databases such as GO Function. A $t$-test was done between eigencoefficients of the genes annotated to a term to any other term in the database. Finally, compared between samples enrichment is calculated by using a Mann–Whitney $U$-test between the $Z$-score of the gene set compared to the $Z$-score of the genes not included in the network. Genes meeting a Bonferroni significance threshold of $P = 9.94E{-}07$ (0.10/100,572) was used. Agnostic analyses of pathways in databases such as Reactome and GO were done to identify pathways relevant to ADHD.

**Phenome-wide association studies**. To identify phenotypes associated with the top eQTL for each TWAS gene, a phenome-wide association study (pheWAS) was done for each SNP. The top three phenotypes (excluding ADHD) were reported. PheWAS was done using public data provided by GWASAtlas (https://atlas.ctglab.nl).

**Genetic correlation**. To determine the genetic relationship between ADHD and the phenotypes identified from pheWAS, genetic correlation of the traits was done for available GWAS data. This was done using GWASAtlas (https://atlas.ctglab.nl), which uses LDSC to determine genetic correlation[24]. The most recent GWAS data (as of 2019) for each trait was used for the correlation[6]. The significance threshold was corrected for the number of tested traits with a Bonferroni correction.

**Reporting summary**. Further information on research design is available in the Nature Research Reporting Summary linked to this article.

## Data availability

Summary statistics have been attached as Supplementary Data 1. All other data are contained within the article or its supplementary information and upon reasonable request.

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

## Acknowledgements

We would like to acknowledge Jay Ross, Cynthia Bourassa, Nargess Farhang, Elias Jabbour, Nadine Nzirorera, and Maryam Tahir for their scientific advice and editing of the manuscript.

## Author contributions

C.L. performed all analyses and drafted the manuscript. D.S. and A.D.L. helped with data management. R.J., P.A.D., and G.A.R. oversaw the manuscript.

## Competing interests

The authors declare no competing interests.
