## [Peer Review File · Nature Communications]

Reviewers' Comments:

Reviewer #1:

Remarks to the Author:

The authors perform an imputed transcriptome association study using summary results from the largest ADHD GWAS to date with 19K cases and 34K controls. They investigate the mechanisms behind ADHD GWAS loci using transcriptome integration approaches such as FUSION and FOCUS. Multiple brain eQTL panels are used including GTEx and CommonMind.

They identified 9 gene with the main approach but finemapping and omnibus test provide a partially overlapping list of candidate genes. Finally, they use the genetic correlation of phewas associated traits of top-eQTLs (of associated genes) to establish a link with educational attainment, ever smoker status, maternal smoking at birth, BMI, and schizophrenia.

The manuscript reports some interesting results but lacks a compelling or unified message. FUSION found 9 or 17 genes in various brain regions. Omnibus test gives another list of genes. Conditional analysis indicates that some of the GWAS loci can be explained with imputed expression. FOCUS (gene finemapping method) prioritizes KAT2B and TMEM161B but other genes have very low probability of causality. So what's the main list of results the authors feel confident reporting? Conditional analysis and finemapping results don't seem to be very consistent. The multiple steps through which education attainment and other phenotypes are linked does not seem very convincing. The methods section lacks some details.

Below are some comments and questions to the authors

Causality claims should be dropped. The causal language in discussion is more careful. "Fine-mapping of TWAS association causally implicates TMEM161B, KAT2B and CTC- 498M16.4" (page 5) To your credit, the discussion section is more careful about not making causality claims.

The rationale behind permuting weights is unclear. There is no reason one would expect that permuting the weights will represent any kind of null.

Claims that Transcriptome imputation increases statistical power is not supported by the results of this study.

The associated phenotypes such as educational attainment and others are linked using a rather lengthy chain of associations. Or at least that's what I interpret from the methods section. This argument has too many steps to be convincing. Genetic correlation try to use large number of small effects likely distributed across multiple sites whereas the phewas phenotypes are linked via a localized association with the topQTL of implicated genes. It would be good to check whether ADHD has direct genetic correlation with the phenotypes reported. If so, the multi-step chain of reasoning seems unnecessary. On the contrary, if the correlation is not found directly with the ADHD GWAS results, the link would sound even less convincing. It may be that you did the latter but from the methods description it is hard to tell exactly.

Methods need more detailed description.

- I could not find where the prediction models came from. Were they downloaded with FUSION?
- which method did you use for the conditional analysis with summary results?
- which method did you use for calculating genetic correlation?

In the abstract, you report that you "successfully demonstrate that several previous GWAS hits can be largely explained by expression". Could you quantify this? How many loci? What proportion of GWAS loci?

Some background of what the original GWAS study found would be helpful to the reader. For

example, what is the current tally of GWAS significant loci. For how many of those do the authors find significant imputed gene association? Were there any genes that were beyond the GWAS loci?

Conditional on predicted gene expression, top SNPs become not significant. This is interesting and would be a good example of increased power by the predicted expression because of multiple independent eQTL signal adding evidence in concert. What did the prediction models look like for these genes? Some discussion on how the multi-SNP prediction explained away the top GWAS SNP would be very helpful.

Reassuringly, a couple of relevant pathways are enriched but as usual with pathway analysis no new insights are obtained.

Brain region hits

Expression regulation tend to be common across tissues. This makes eQTL based imputed Transcriptome methods not very effective in identifying "causal" tissues or regions. There are several instances in the manuscript, where you find a significant gene in a tissue you consider relevant and link it to some finding about the tissue. This argument as a form of validation is not very convincing when mostly relevant tissues were selected to start with. You were bound to find associations in relevant tissues. The identification of a brain region would be relevant if the association was specific to that region. It is unclear from the manuscript that this was the case.

Could you add a discussion on consistency across eQTL panels and brain regions?.

Minor comments

"We successfully demonstrate that several previous GWAS hits can be largely explained by expression."

should be ... "by expression regulation" or something along those lines.

Please add line and page numbers

page 8: "Future studies should investigate segregating large heterogeneous cohorts, "
This sentence seems to be a bit disconnected with the rest of the paragraph.

Reference 7 and 9 are the same.

Unclear what you mean by "genes coexpressed with genetic predictor"

Unfinished sentence

"Interestingly, there was a strong negative correlation between educational attainment (Supplementary Figure 1)."

between education attainment and what?

Or perhaps you meant "with" instead of "between"

correct this sentence

"a genetic correlation for risky behaviours such as "ever-smoker" and maternal smoking around birth were positively correlated."

Reviewer #2:

Remarks to the Author:

Referee report for NCOMMS-19-17071-T

Title: Transcriptome-wide association study of attention deficit hyperactivity disorder identifies associated genes and phenotypes

The authors have conducted a transcriptome-wide association study (TWAS) of ADHD based on the largest GWAS of ADHD to date. They identify 9 transcriptome-wide significant hits and demonstrate that several of the GWAS hits previously found can be explained by expression data. They continue the analyses with probabilistic causal fine mapping tools and pathway enrichment highlighting more genes and implicating dopaminergic and norepinephrine pathways.

This is a natural follow-up to the latest and largest GWAS of ADHD and the results should be of interest to the broader psychiatric genetics community as well as researchers of ADHD specifically. The methodology used is sound and it by and large follows the best practices that has been carefully thought out and presented in a recent Nature Genetics paper by Wainberg et al (2019). It is the first time I have seen a study of ADHD reported following that process. However, there are publications that employ similar methods to the same ADHD GWAS and the same or related expression sources:

Fahira et al, "Prediction of causal genes and gene expression analysis of attention-deficit hyperactivity disorder in the different brain region, a comprehensive integrative analysis of ADHD", Behavioural Brain Research 364 (2019) 183–192: Use GTEx as well as other eQTL data sets, but not CommonMind as in the present manuscript. They employ different, but closely related methodology and also arrive at partially overlapping results.

Gamazon et al, "Multi-tissue transcriptome analyses identify genetic mechanisms underlying neuropsychiatric traits", Nature Genetics 51 (2019) 933–940: Use the same data but somewhat different methods, and as they deal with more phenotypes does not go the same depth with respect to ADHD.

This does subtract a bit from the novelty of the work. On the other hand there is value in interrogating the same data from slightly different angles. This is particularly true when the road from raw data to end result is as long and convoluted as is the case with these sophisticated and very involved bioinformatics procedures.

Overall, the paper is well written and kept nice and brief, In case of the methods a bit too brief as it would be hard to reproduce the analyses based on the description given. In particular:

1. P5, last par. - p6, 1st par. and P10, 2nd par.: It is unclear how exactly the gene-set analyses were conducted. Please elaborate: What tool was used, what statistical method was employed. What metric and method was the clustering based on, and what role did the clustering play in the enrichment analysis? Also, state the justification for the nominal significance level of 0.30.

2. P9, 2nd par.: When conditioning in the joint gene model, how are the regions on which you condition defined - are you using the transcribed region of the genes or do you include flanking sequences? Please also write up the model or as a minimum cite the tool used (I assume it is also FUSION that was used here).

With respect to reporting results:

3. It would be helpful to have a table providing an overview of the loci/genes implicated in the GWAS that forms the basis for this analysis and the loci/genes implicated here, showing what genes from the GWAS that harvested more evidence here, which that did not, and what new loci were detected here that were missed by the GWAS. Table one could be expanded like that; preferably removing the genes with "suggestive significance".

The rest are really minor comments:

4. P3, 1st par.: Change "...more interpretable biologically-relevant results due their use..." to "...more interpretable biologically-relevant results due to their use...?"
5. P4, last par.: References to Figure 2A and 2B have been swapped and so has 2C and 2D — this also means that figures do not appear in the order they are mentioned. Besides letters a, b, c, d and e are capitalized in the figures but not when referencing them.
6. P5, last par.: Change "The genes grouped into three different clusters based on co-expression of public RNA-seq data" to "The genes were grouped into three different clusters based on co-expression of public RNA-seq data"
7. P6, 2nd par.: "Since most eQTLs were associated with ADHD, we chose to exclude it from Table 3 to emphasize the other three top phenotypes per SNP": Refer simply to "the table" here or write it out as "Supplementary Table 3". Simply writing "Table 3" is confusing.
8. P6, 2nd par.: What is meant by "Several risk-associated phenotypes such as "ever-smoker", alcohol intake over 10 years, and maternal smoking around birth..."? That they are risk factors for ADHD? - State that more clearly.
9. P6, 3rd par.: All these genetic correlations except for that for maternal smoking around birth were reported in Demontis et al 2019. Also, please provide citations for the papers corresponding to the summary statistics used. "Most recent" may be a relative term depending on your level of access, if you are using published, unpublished summary statistics etc.
10. P10, 1st par.: "The same TWAS reference panels for FUSION were used." - "as in the analysis described above"?
11. Table 1 and 2: The caption should not read "Significant TWAS genes for ADHD" or "Omnibus significant TWAS genes for ADHD" when they also include genes with merely "suggestive significance". Remove the latter or call it "Top ranking genes in the TWAS" or something to that effect.
12. Table 1-3: Please limit the number of decimal places. The number of decimals shown by far exceeds what precision these methods provide.
13. Captions for Figures 2x are identical. Please customize the text to explain each figure individually. No need to state genes marginally associated are coloured blue, when there are none. Better to state that no genes were marginally associated. Moreover, it is easy to miss one in the cloud of genes in the plot, it would be helpful to the reader to explicitly write the marginally associated genes in the caption as well as the genes on which you are conditioning.
14. Figures 2x: The colours green and blue are hard to do distinguish for small genes, so consider choosing colours that are easier to separate.
15. Figures 2B and 2C: Some genes seem to be duplicated or triplicated: PIDD, AP006621.6 and AP006621.5 in 2B, and ARTN in 2C, where there are 2 blue copies are together and a green copy further down. There can obviously be a point in showing genes significant in both GWAS and TWAS this way, but that cannot explain all the duplications and triplications. If there is a meaning to it, explain in the caption. If it is a mistake, it is best to fix it even if it is innocents.
16. Suppl. Figure 2: The edges of the graph as very hard to see. What is more, if the width are to indicate the weight of the connection, it is hard to understand the clustering.

Response to reviewers

Reviewer #1 (Remarks to the Author):

The authors perform an imputed transcriptome association study using summary results from the largest ADHD GWAS to date with 19K cases and 34K controls. They investigate the mechanisms behind ADHD GWAS loci using transcriptome integration approaches such as FUSION and FOCUS. Multiple brain eQTL panels are used including GTEx and CommonMind.

They identified 9 gene with the main approach but finemapping and omnibus test provide a partially overlapping list of candidate genes. Finally, they use the genetic correlation of phewas associated traits of top-eQTLs (of associated genes) to establish a link with educational attainment, ever smoker status, maternal smoking at birth, BMI, and schizophrenia.

The manuscript reports some interesting results but lacks a compelling or unified message. FUSION found 9 or 17 genes in various brain regions. Omnibus test gives another list of genes. Conditional analysis indicates that some of the GWAS loci can be explained with imputed expression. FOCUS (gene finemapping method) prioritizes KAT2B and TMEM161B but other genes have very low probability of causality. So what's the main list of results the authors feel confident reporting? Conditional analysis and finemapping results don't seem to be very consistent. The multiple steps through which education attainment and other phenotypes are linked does not seem very convincing. The methods section lacks some details.

Below are some comments and questions to the authors:

[Author's response] We would like to thank the reviewer for these constructive comments. We believe that our answers will contribute to significantly improved our manuscript.

In regard to the method section we have now modified the discussion to better highlight what is concordant between the methods used: FUSION's TWAS, omnibus, and fine-mapping. For instance, the lncRNA gene *CTC-498M16.4* gene is amongst the top prioritized hits across all three tests (**line 244-246**).

[Query 1] Causality claims should be dropped. The causal language in discussion is more careful. "Fine-mapping of TWAS association causally implicates *TMEM161B*, *KAT2B* and *CTC-498M16.4*" (page 5) To your credit, the discussion section is more careful about not making causality claims.

[Author's response to Q1] We have corrected this sentence to "evidence for causality" instead of claims of causality. This statement is meant to pertain the probabilistic fine-mapping of TWAS associated genes and those included within the 90%-confidence causal gene set were listed in the Table; hence these genes were prioritized for their potential causality.

[Query 2] The rationale behind permuting weights is unclear. There is no reason one would expect that permuting the weights will represent any kind of null.

[Author's response to Q2] FUSION's permutation step seeks to determine the significance of a trait-expression association conditional on the trait-SNP effects at the

locus. The decision to use this null allowed us to determine how much signal is derived from expression (given the GWAS signals at a specific locus and not due to chance colocalization (**lines 313-314**)). The original report (Gusev et al., *Integrative approaches for large-scale transcriptome-wide association studies*, 2016) describing this form of analysis also used a permutation step; something which has also been described in more recent TWAS reports (e.g. Mancuso et al., *Large-scale transcriptome-wide association study identifies new prostate cancer risk regions*, 2018). Gusev et al., *Transcriptome-wide association study of schizophrenia and chromatin activity yields mechanistic disease insights*, 2018).

[Query 3] Claims that Transcriptome imputation increases statistical power is not supported by the results of this study.

[Author's response to Q3] We concur that the phrasing used in the submitted manuscript could be misleading. What we meant to emphasize was that some of the top signals identified using a TWAS approach were never observed in studies where a classical GWA approach was used. We have modified the text as follow: “*Ultimately, TI provides the opportunity to increase the ability to detect putative genes with small effect sizes that are associated with a disease.*” (**line 91-92**).

[Query 4] The associated phenotypes such as educational attainment and others are linked using a rather lengthy chain of associations. Or at least that's what I interpret from the methods section. This argument has too many steps to be convincing. Genetic correlation try to use large number of small effects likely distributed across multiple sites whereas the phewas phenotypes are linked via a localized association with the topeQTL of implicated genes. It would be good to check whether ADHD has direct genetic correlation with the phenotypes reported. If so, the multi-step chain of reasoning seems unnecessary. On the contrary, if the correlation is not found directly with the ADHD GWAS results, the link would sound even less convincing. It may be that you did the latter but from the methods description it is hard to tell exactly.

[Author's response to Q4] We agree and edited the manuscript to clarify what we meant to convey, which is that these can re-affirm phenotypes previously implicated in ADHD and reveal novel observation. Our manuscript highlighted phenotypes which were not previously associated with ADHD (at a genetic level) but which were previously implicated as environmental risk factors (e.g. maternal smoking; **lines 181-183**).

[Query 5] Methods need more detailed description. - I could not find where the prediction models came from.

*Were they downloaded with FUSION?

[Author's response] Yes they were. We've added this statement to make it clearer. Please see **line 292**.

*Which method did you use for the conditional analysis with summary results?

[Author's response] We used FUSION's conditional analysis option. This has been clarified this in the revised manuscript (**line 308**).

*Which method did you use for calculating genetic correlation?

[Author's response] LDSC was used. This has been added and cited in the revised manuscript (**line 356**).

[Query 6] In the abstract, you report that you "successfully demonstrate that several previous GWAS hits can be largely explained by expression". Could you quantify this? How many loci? What proportion of GWAS loci?

[Author's response to Q6] We quantified it and have now included this information in the abstract. We highlighted that six genes were not implicated in the original GWAS. Two of these GWAS signals are explained by the Bonferroni-corrected TWAS hits (**lines 42-43**).

[Query 7] Some background of what the original GWAS study found would be helpful to the reader. For example, what is the current tally of GWAS significant loci. For how many of those do the authors find significant imputed gene association? Were there any genes that were beyond the GWAS loci?

[Author's response to Q7] We have emphasized what the number of GWAS loci for ADHD are (line 75-76). We have also highlighted (Table 1) how many of those loci are from previous GWA studies and how many were not previously reported.

[Query 8] Conditional on predicted gene expression, top SNPs become not significant. This is interesting and would be a good example of increased power by the predicted expression because of multiple independent eQTL signal adding evidence in concert. What did the prediction model look like for these genes? Some discussion on how the multi-SNP prediction explained away the top GWAS SNP would be very helpful.

[Author's response to Q8] We agree with the reviewer and have discussed the model further in the results (**lines 136-139**) and discussion (**lines 207-212**) sections.

[Query 9] Brain region hits. Expression regulation tend to be common across tissues. This makes eQTL based imputed Transcriptome methods not very effective in identifying "causal" tissues or regions. There are several instances in the manuscript, where you find a significant gene in a tissue you consider relevant and link it to some finding about the tissue. This argument as a form of validation is not very convincing when mostly relevant tissues were selected to start with. You were bound to find associations in relevant tissues. The identification of a brain region would be relevant if the association was specific to that region. It is unclear from the manuscript that this was the case. Could you add a discussion on consistency across eQTL panels and brain regions?

[Author's response to Q9] We have expanded this in the paper and highlighted that certain significant hits were specific to specific tissue types. We have also pointed out interpretation drawbacks, such that this may also be due to differences in RNA quality and panel size (**lines 220-228**).

Minor comments

[Query 10] We successfully demonstrate that several previous GWAS hits can be largely explained by expression." should be ... "by expression regulation" or something along those lines.

[Author's response to Q10] This has been fixed.

[Query 11] Please add line and page numbers

[Author's response to Q11] We have done this. Our apologies for the inconvenience.

[Query 12] page 8: "Future studies should investigate segregating large heterogeneous cohorts, "This sentence seems to be a bit disconnected with the rest of the paragraph.

[Author's response to Q12] We have removed this sentence from the revised manuscript as we concur that it was confusing.

[Query 13] Reference 7 and 9 are the same.

[Author's response to Q13] This has been corrected.

[Query 14] Unclear what you mean by "genes coexpressed with genetic predictor"

[Author's response to Q14] We have removed this sentence as it was incorrectly worded.

[Query 15] This would be an unfinished sentence: "Interestingly, there was a strong negative correlation between educational attainment (Supplementary Figure 1)."between education attainment and what? Or perhaps you meant "with" instead of "between".

[Author's response to Q15] Indeed we meant "with" instead of "between". We have fixed it to the following sentence: Interestingly, there was a strong negative correlation between educational attainment and ADHD (**line 189-190**).

[Query 16] Correct this sentence "a genetic correlation for risky behaviours such as "ever-smoker" and maternal smoking around birth were positively correlated."

[Author's response to Q16] We have revised the sentence as follow: Furthermore, there was a positive correlation with maternal smoking around birth, body mass index, ever smoker, and schizophrenia (**lines 190-193**).

Reviewer #2 (Remarks to the Author):

The authors have conducted a transcriptome-wide association study (TWAS) of ADHD based on the largest GWAS of ADHD to date. They identify 9 transcriptome-wide significant hits and demonstrate that several of the GWAS hits previously found can be explained by expression data. They continue the analyses with probabilistic causal fine mapping tools and pathway enrichment highlighting more genes and implicating dopaminergic and norepinephrine pathways.

This is a natural follow-up to the latest and largest GWAS of ADHD and the results should be of interest to the broader psychiatric genetics community as well as researchers of ADHD specifically. The methodology used is sound and it by and large follows the best practices that has been carefully thought out and presented in a recent Nature Genetics paper by Wainberg et al (2019). It is the first time I have seen a study of ADHD reported following that process. However, there are publications that employ similar methods to the same ADHD GWAS and the same or related expression sources:

Fahira et al, “Prediction of causal genes and gene expression analysis of attention-deficit hyperactivity disorder in the different brain region, a comprehensive integrative analysis of ADHD”, Behavioural Brain Research 364 (2019) 183–192: Use GTEx as well as other eQTL data sets, but not CommonMind as in the present manuscript. They employ different, but closely related methodology and also arrive at partially overlapping results.

Gamazon et al, “Multi-tissue transcriptome analyses identify genetic mechanisms underlying neuropsychiatric traits”, Nature Genetics 51 (2019) 933–940: Use the same data but somewhat different methods, and as they deal with more phenotypes does not go the same depth with respect to ADHD.

This does subtract a bit from the novelty of the work. On the other hand there is value in interrogating the same data from slightly different angles. This is particularly true when the road from raw data to end result is as long and convoluted as is the case with these sophisticated and very involved bioinformatics procedures.

[Author’s response] Thank you for your comments and suggestions to improve the manuscript.

Overall, the paper is well written and kept nice and brief, In case of the methods a bit too brief as it would be hard to reproduce the analyses based on the description given. In particular:

[Query 1] 1. P5, last par. - p6, 1st par. and P10, 2nd par.: It is unclear how exactly the gene-set analyses were conducted. Please elaborate: What tool was used, what statistical method was employed. What metric and method was the clustering based on, and what role did the clustering play in the enrichment analysis? Also, state the justification for the nominal significance level of 0.30.

[Author’s response to Q1] For the statistical issue we have now added this information (**lines 333-344**). We have also expanded the text detailing the purpose of using the gene networking / clustering (**lines 248-254**). Briefly, this section of the manuscript highlights the concordance of the TWAS results (i.e. *ELOVL1*, *TIE1* and *MED8* which are all significant in the dorsolateral prefrontal cortex and clustered together). Similarly, *CCDC24*, *ARTN* and *MANBA* which are all genes relevant to the cerebellum and clustered together. Hence, we wish to highlight the relevance that these 6 hits are all from the same locus, but are clustered separately in the gene network and specific to different

brain tissue types, suggesting tissue-specificity and that a single locus can affect tissues differently.

[Query 2] 2. P9, 2nd par.: When conditioning in the joint gene model, how are the regions on which you condition defined - are you using the transcribed region of the genes or do you include flanking sequences? Please also write up the model or as a minimum cite the tool used (I assume it is also FUSION that was used here).

[Author's response to Q2] We have now cited the tool (FUSION) and explained it in the text (**line 308**).

[Query 3] With respect to reporting results: It would be helpful to have a table providing an overview of the loci/genes implicated in the GWAS that forms the basis for this analysis and the loci/genes implicated here, showing what genes from the GWAS that harvested more evidence here, which that did not, and what new loci were detected here that were missed by the GWAS. Table one could be expanded like that; preferably removing the genes with “suggestive significance”.

[Author's response to Q3] We agree and have now expanded Table 1 to include these suggestions (**page 12-13**).

Minor comments

[Query 4] P3, 1st par.: Change “...more interpretable biologically-relevant results due their use...” to “...more interpretable biologically-relevant results due to their use...”?

[Author's response to Q4] We've have corrected this.

[Query 5] P4, last par.: References to Figure 2A and 2B have been swapped and so has 2C and 2D — this also means that figures do not appear in the order they are mentioned. Besides letters a, b, c, d and e are capitalized in the figures but not when referencing them.

[Author's response to Q5] Thank you for pointing out this error. We realized a mix-up occurred when combining files. We have now corrected this.

[Query 6] P5, last par.: Change “The genes grouped into three different clusters based on co-expression of public RNA-seq data” to “The genes were grouped into three different clusters based on co-expression of public RNA-seq data”

[Author's response to Q6] This has been corrected.

[Query 7] P6, 2nd par.: “Since most eQTLs were associated with ADHD, we chose to exclude it from Table 3 to emphasize the other three top phenotypes per SNP”: Refer simply to “the table” here or write it out as “Supplementary Table 3”. Simply writing “Table 3” is confusing.

[Author's response to Q7] This has been corrected.

[Query 8] P6, 2nd par.: What is meant by “Several risk-associated phenotypes such as “ever-smoker”, alcohol intake over 10 years, and maternal smoking around birth...”? That they are risk factors for ADHD? - State that more clearly.

[Author's response to Q8] We've just added a statement clarifying this in lines 208-210.

[Query 9] P6, 3rd par.: All these genetic correlations except for that for maternal smoking around birth were reported in Demontis et al 2019. Also, please provide citations for the papers corresponding to the summary statistics used. “Most recent” may be a relative term depending on your level of access, if you are using published, unpublished summary statistics etc.

[Author’s response to Q9] This has been added.

[Query 10] P10, 1st par.: “The same TWAS reference panels for FUSION were used.” - “as in the analysis described above”?

[Author’s response to Q10] Yes. This has been clarified.

[Query 11] Table 1 and 2: The caption should not read “Significant TWAS genes for ADHD” or “Omnibus significant TWAS genes for ADHD” when they also include genes with merely “suggestive significance”. Remove the latter or call it “Top ranking genes in the TWAS” or something to that effect.

[Author’s response to Q11] We have implemented these corrections.

[Query 12] Table 1-3: Please limit the number of decimal places. The number of decimals shown by far exceeds what precision these methods provide.

[Author’s response to Q12] This has been corrected to two decimal places for all numbers.

[Query 13] Captions for Figures 2x are identical. Please customize the text to explain each figure individually. No need to state genes marginally associated are coloured blue, when there are none. Better to state that no genes were marginally associated. Moreover, it is easy to miss one in the cloud of genes in the plot, it would be helpful to the reader to explicitly write the marginally associated genes in the caption as well as the genes on which you are conditioning.

[Author’s response to Q13] Thank you for the suggestion. The text has been customized.

[Query 14] Figures 2x: The colours green and blue are hard to do distinguish for small genes, so consider choosing colours that are easier to separate.

[Author’s response to Q14] We have changed it to orange and green to better differentiate.

[Query 15] Figures 2B and 2C: Some genes seem to be duplicated or triplicated: PIDD, AP006621.6 and AP006621.5 in 2B, and ARTN in 2C, where there are 2 blue copies are together and a green copy further down. There can obviously be a point in showing genes significant in both GWAS and TWAS this way, but that cannot explain all the duplications and triplications. If there is a meaning to it, explain in the caption. If it is a mistake, it is best to fix it even if it is innocents.

[Author’s response to Q15] We have fixed the issue of duplications and realized it was an error in an input file.

[Query 16] Suppl. Figure 2: The edges of the graph as very hard to see. What is more, if the width are to indicate the weight of the connection, it is hard to understand the clustering.

[Author's response to Q16] Yes, it indicates the weight of connection. We have clarified this in the figure legend and also darkened the lines slightly to make it easier to see.

Reviewers' Comments:

Reviewer #1:

Remarks to the Author:

The authors have addressed all substantive issues.

Reviewer #2:

Remarks to the Author:

Overall, the authors have responded satisfactorily to nearly all comments. However, a couple of things are still lacking:

In my review I pointed out two publications employing very similar methods to the same data sources, but argued that although this might subtract from the novelty, there is still value in interrogation of the data from different angles. I however failed to make it clear that this value only really come to fruit if the authors actually relates to these other analyses in the manuscript. The paper would therefore improve significantly by including a discussion of these other approaches and what new we can learn from these additional analyses.

Q1: The gene-set analysis is now explained, but there is still no justification for the apparently arbitrary choice of significance level of 0.3. Yes, the dependence means that Bonferroni is conservative, are there reasons to believe the 'effective number of tests' is exactly one sixth of the total number of tests? Please provide the arguments that led to this number, or consider if some sort of resampling or permutation test may be deployed to deal with this issue.

One minor point: I328--330: 'A relaxed Bonferroni-corrected p-value of <0.30 (uncorrected $p < 2.98E-06$) was used because co-regulation in TWAS signals violate the independence assumption required for Bonferroni-correction, making it too strict, especially for gene-set enrichment.' There seem to be some confusion of p-value and p-value threshold, alpha-level.

Reviewers' comments:

Reviewer #1 (Remarks to the Author):

The authors have addressed all substantive issues.

[Response to reviewers]: We would like to thank both reviewers for their contributions towards this manuscript.

Reviewer #2 (Remarks to the Author):

Overall, the authors have responded satisfactorily to nearly all comments. However, a couple of things are still lacking:

In my review I pointed out two publications employing very similar methods to the same data sources, but argued that although this might subtract from the novelty, there is still value in interrogation of the data from different angles. I however failed to make it clear that this value only really come to fruit if the authors actually relates to these other analyses in the manuscript. The paper would therefore improve significantly by including a discussion of these other approaches and what new we can learn from these additional analyses.

[Response to reviewers]: Thank you to this reviewer for their helpful suggestions. We have added these references and discussed it further (lines 267-276).

Q1: The gene-set analysis is now explained, but there is still no justification for the apparently arbitrary choice of significance level of 0.3. Yes, the dependence means that Bonferroni is conservative, are there reasons to believe the ``effective number of tests' is exactly one sixth of the total number of tests? Please provide the arguments that led to this number or consider if some sort of resampling or permutation test may be deployed to deal with this issue.

[Response to reviewers]: We have since changed this to 0.10, which is a more commonly accepted nominal significance threshold. We've additionally updated any p-values that have updated the enriched pathways to reflect this and our supplementary figures (Supplementary file).

One minor point: l328--330: ``A relaxed Bonferroni-corrected p-value of <0.30 (uncorrected $p < 2.98E-06$) was used because co-regulation in TWAS signals violate the independence assumption required for Bonferroni-correction, making it too strict, especially for gene-set enrichment.' There seem to be some confusion of p-value and p-value threshold, alpha-level.

[Response to reviewers]: Thank you. We have addressed this issue and changed it to the following "A relaxed Bonferroni-corrected nominal threshold of 0.10 (uncorrected $2.98E-06$) was used because co-regulation in TWAS signals violate the independence assumption required for Bonferroni-correction, making it too strict, especially for gene-set enrichment.'

Reviewers' Comments:

Reviewer #2:

Remarks to the Author:

The authors have addressed all the remaining issues.